# Postmortem Metabolism and Pork Quality Development Are Affected by Electrical Stimulation across Three Genetic Lines

**DOI:** 10.3390/ani13162599

**Published:** 2023-08-11

**Authors:** Matthew D. Spires, Jocelyn S. Bodmer, Mariane Beline, Jordan C. Wicks, Morgan D. Zumbaugh, Tim Hao Shi, Brian T. Reichert, Allan P. Schinckel, Alan L. Grant, David E. Gerrard

**Affiliations:** 1Department of Animal Sciences, Purdue University, West Lafayette, IN 47907, USA; ambermatthew@gmail.com (M.D.S.); brichert@purdue.edu (B.T.R.); aschinck@purdue.edu (A.P.S.); 2School of Animal and Sciences, Virginia Polytechnic Institute and State University, Blacksburg, VA 24061, USA; jocelynb@vt.edu (J.S.B.); mbeline@vt.edu (M.B.); jcwicks@vt.edu (J.C.W.); haoshi@vt.edu (T.H.S.); algrant@vt.edu (A.L.G.); 3Department of Animal Sciences and Industry, Kansas State University, Manhattan, KS 66506, USA; mdzumbaugh@ksu.edu

**Keywords:** genetics, glycolysis, glycogen, pH, pork quality

## Abstract

**Simple Summary:**

Fresh pork quality is an important discriminator for consumers making purchasing decisions. As such, this issue deserves significant attention from all sectors of the swine industry. Development of pork quality is a complex process predicated largely on the metabolism that occurs in the muscle after death. Mechanisms driving this biochemistry are not well established but are sensitive to pig genetic line and postmortem handling procedures. We attempted to unravel some of these processes by subjecting three different breeds of pigs to electrical stimulation postmortem in an effort to simulate adverse handling events after stunning. We then described the muscle characteristics of these lines and monitored metabolite accumulation in the tissue over 24 h and related it to ultimate pork quality. Differences in myosin heavy chain (MyHC) and metabolism were detected across genetic lines, but pork quality was generally unaffected. These data show different genetic lines can impact postmortem metabolism but a greater understanding of mechanisms controlling the transformation of muscle to meat is necessary before we can select for tissue parameters to improve fresh pork quality development and consistency.

**Abstract:**

Variations in postmortem metabolism in muscle impact pork quality development. Curiously, some genetic lines are more refractile to adverse pork quality development than others and may regulate energy metabolism differently. The aim of this study was to challenge pork carcasses from different genetic populations with electrical stimulation (ES) to determine how postmortem metabolism varies with genetic line and explore control points that reside in glycolysis in dying muscle. Three genetic populations (GP) were subjected to ES (100 V or 200 V, 13 pulses, 2 s on/2 s off) at 15- or 25-min post-exsanguination, or no stimulation (NS). Genetic population affected relative muscle relative abundance of different myosin heavy chains, glycogen, G6P, and lactate concentrations. Genetic lines responded similarly to ES, but a comparison of ES treatment groups revealed a trend for an interaction between voltage, time of ES, and time postmortem. Higher voltage accelerated pH decline at 20 min up to 60 min postmortem. Trends in color and firmness scores and L* values were consistent with pH and metabolite data. These data show that genetic populations respond differently to postmortem perturbation by altering glycolytic flux and suggest differences in postmortem glycolysis may be partially responsible for differences in meat quality between genetic populations, though not entirely.

## 1. Introduction

The pork industry has made significant progress in meeting the ever-changing demands of a global citizenry. For nearly 30 years, pork producers and genetic companies have selected pigs that produce leaner, heavier-muscled carcasses. Part of this is driven by growing concerns over dietary fat intake by consumers [1]. However, other reasons are related to increasing production efficiencies, where larger-framed, later-maturing market hogs reach heavy weights through protracted lean growth curves. These heavier weights are favored by most processors because they dilute overhead and labor costs [2]. These pressures have led to the evolution of multiple genetic lines with phenomenal abilities to grow and deposit lean tissue [3]. Unfortunately, pork quality has languished with changes in carcass composition, which are further codified within genetic lines [2]. Variations in fresh pork quality characteristics negatively impact consumer perception, and this is not without merit, as visible fresh meat quality attributes closely align with eating quality outcomes and processing performance of fresh pork. Bowker et al. [4] observed that pork quality traits exhibited by emerging pig genetic lines exhibit resemblances, either in part or in whole, to those prevailing in the swine industry during the late 1960s and early 1970s. Modern lines have less intramuscular fat, higher moisture content and result in products that resemble pale, soft, and exudative (PSE) pork. Pale, soft, exudative pork is one of the most severe quality defects affecting a consumer’s decision to purchase fresh and processed pork. This defect is accompanied by a considerable loss in water-holding capacity and is subject to increased purge during refrigerated storage, costing the meat industry millions in lost revenues [5,6]. Product losses due to purge can average 1–3% in fresh retail cuts [7], while PSE meat products can be as high as 10% [8]. In addition, due to the lack of water-holding capacity, once cooked the pork product will be dry with an unsatisfactory texture, resulting in a poor eating experience for the consumer.

Considerable efforts have been expended to understand the mechanisms responsible for adverse pork quality development. Pale, soft, and exudative pork was first coined for lean from pigs possessing a naturally occurring mutation in the ryanodine receptor (calcium gating) or a positive reactor to halothane (HAL) gas. Muscles from these pigs experience an accelerated metabolism postmortem with elevated carcass temperatures, resulting in a faster rate of pH decline and low ultimate pH. This combination of low pH and high temperature results in excessive protein denaturation and a product with impaired water-binding ability and color, quite an unsavory sight. The other major meat quality defect results in similar quality characteristics to PSE meat yet arises from a lower than normal ultimate pH in the muscle (pH < 5.4) and is sometimes called acid meat or the Hampshire effect [9,10,11]. The biochemical mechanisms responsible for breaching the ultimate pH is a bit vague, but mitochondria have been implicated in mediating this process [12].

Despite selection against the RN and HAL genes, there is still a relatively high occurrence of PSE pork [5,13]. Efforts to establish models to study PSE meat development have been attempted, such as holding carcasses at elevated temperatures (37 °C) until the onset of rigor causes meat to become PSE [14,15,16]. Additionally, others have used electrical stimulation (ES) as a means of accelerating postmortem muscle metabolism and causing PSE meat [17,18]. The ES approach of producing PSE meat produces muscle pH and temperature declines that closely resemble those of animals that develop PSE naturally [17]. Therefore, to explore the mechanisms controlling pork quality, we challenged three genetic populations of pigs, all RN- and HAL-negative with ES, to explore the impact of genetic line on pork quality development and investigate what steps in glycolysis regulate this process. The rationale is that if we can identify the precise biochemical step(s) responsible for controlling postmortem metabolism in muscle, then we will be better positioned to either: (1) modify the process; (2) develop technologies to predict and segregate pork quality development ‘on-line’; and (or) (3) select against pigs “vulnerable” to adverse quality development.

## 2. Materials and Methods

### 2.1. The Animals and Experimental Design

One hundred and fifty castrated males from three genetic populations (GP, n = 50/GP), were obtained at 14–17 days of age and fed a commercial diet ad libitum to market weight (115 ± 2 kg) at the Purdue University Animal Science Research and Education Center. All procedures were approved by the Purdue University Animal Care and Use committee (PACUC #98-107). Animals were housed in a three double-L fiberglass barns equipped with 4′ × 4′ pens (5 pigs/pen). Lines consisted of terminal cross (Lean European terminal sire line) sires mated to Landrace (Duroc–Large White) females (GP1). GP1 animals contain the highest percent lean (heavy-muscled). The second line consisted of Duroc–Large White sires mated to Yorkshire–Landrace females (GP2). GP2 animals are moderately muscled animals. The third line consisted of US Duroc sires mated to Yorkshire by Duroc–Landrace dams (GP3). GP3 animals had the lowest percent lean (light muscled). A characterization of GPs performance can be observed in Table 1. All animals used in this study were RN- and HAL-negative.

At slaughter, 20 animals of each GP were randomly selected and assigned to a completely randomized design in a 3 × 2 × 2 factorial arrangement. Factors consisted of three genetic populations (GP1, GP2, and GP3), two electrical stimulation (ES) voltages (100 V and 200 V), and two ES times postmortem (15 min and 25 min). Animals were harvested at the Purdue University Meat Science Research and Teaching Laboratory. All pigs were immobilized by electrical stunning and exsanguinated according to standard harvesting procedures. Following evisceration, the ES was applied, and carcasses were placed in a cooler (4 °C). ES was administered via a 16.5 cm long electrical probe inserted in the muscle of the left shoulder, with 13 pulses of two seconds duration and 2 s pause interval.

### 2.2. Muscle Sampling

Samples were collected from the Longissimus lumborum (LL) of each carcass at 0, 30, 60, and 1440 min postmortem for the determination of metabolite concentrations. Samples were obtained using a 1.3 cm coring tool and were immediately frozen in liquid nitrogen and stored at −80 °C. After the chilling period (4 °C), carcasses were split and ribbed between the 10th and 11th ribs and were subjectively evaluated for color and firmness as proposed by NPPC [19]. Briefly, after a 10 min bloom time, samples were classified based on a six-point color scale ranging from 1–6 (pale pinkish gray to white to dark purplish red, respectively). Firmness was assessed after color evaluation, with samples being classified on a three-point scale of 1 (soft), 2 (firm) and 3 (very firm). Also, two 2.5-cm-thick chops were removed from the tenth rib region for the determination of water-holding capacity and objective color.

### 2.3. Temperature and pH

Muscle pH was measured at 1, 10, 20, 30, 40, 50, 60, and 1440-min post-exsanguination from the LL using a Beckman 110 ISFET pH meter with a spear-tipped KCl-gel probe (Fullerton, CA, USA). Muscle temperature was obtained at 1, 10, 20, 30, 40, 50, 60 min post-exsanguination, approximately 2 cm caudal from the point of pH measurement, at a depth of approximately 4–5 cm, using a VWR brandTM T Digital Thermometer (Friendswood, TX, USA).

### 2.4. Color and Water-Holding Capacity

One of the 2.5 cm LL chops was used for objective color determination using the CIELab system [20], utilizing a Hunter Lab 45/0 D25-PC2 Colorimeter (Hunter and Associates Laboratory Inc., Reston, VA, USA). L* (luminosity), a* (redness), and b* (yellowness) values of each sample were obtained through an average of three random readings.

The drip loss (DL) method [21] was used to determine water-holding capacity. Three subsamples of the second chop were obtained using a 2.5 cm coring device. Samples were weighed and placed in pre-weighed drip loss tubes. After 24 h at 4 °C, the samples, tubes, and exudate weights were recorded. Drip loss was determined in triplicate as a percentage of moisture loss.

### 2.5. Metabolite Analysis

Frozen muscle samples (0.25–0.3 g) were powdered in liquid nitrogen and homogenized with 10 mL of 0.5 M perchloric acid (PCA). Aliquots of 500 µL and 250 µL each were taken from the homogenate and placed in 1.5 mL microfuge tubes. The 500 µL aliquot was centrifuged at 1500× *g* for 20 min at 4 °C, and supernatants were used for the determination of glucose, glucose-6-phosphate (G6P), and lactate concentrations. The 250 µL aliquots were subjected to hydrolysis with 25 µL of 30% (*v*/*v*) KOH solution and 500 µL of amyl glucosidase (13.0 mg/mL; Sigma Chemical Company, St. Louis, MO, USA) and incubation in a 37 °C agitating water bath for 3 h. Following incubation, 50 µL of 3 M PCA was added, and samples were placed on ice for 10 min. Aliquots were then centrifuged at 1500× *g* for 15 min at 4 °C, and supernatants were used for the determination of glycogen concentration. Glycogen, G6P, glucose, and lactate concentrations were obtained using the enzymatic analytical methods proposed by Bergmeyer [22]. The glycolytic potential was calculated as [Lactate] + 2([Glycogen] + [G6P] + [Glucose]) as proposed by Monin and Sellier [10].

### 2.6. Myosin Heavy Chain Isoform Abundance Analysis

#### 2.6.1. Myosin Extraction

Muscle samples were subjected to myosin extraction as described in [23]. Samples randomly selected from 10 animals of each genetic population were ground in liquid nitrogen, and approximately 1 g was diluted 1:7 (*w*/*v*) in extraction buffer, agitated horizontally for 15 min on ice, and then centrifuged at 10,000× *g* for 20 min at 4 °C. The supernatant was removed, mixed 1:1 (*v*/*v*) with glycerol, and stored at −20 °C. Samples were analyzed for protein content using a BCA assay (Sigma, St. Louis, MO, USA).

#### 2.6.2. ELISA Protocol

Samples were analyzed with an enzyme-linked immunosorbent assay (ELISA), which determines the relative amounts of each myosin heavy chain isoform, I, IIA, and IIB, as described by Depreux et al. [24]. Samples were diluted to 7 μg/mL in a binding buffer (0.05 M Tris HCL) and aliquoted to 96-well plates in five replications and incubated at room temperature overnight to allow the myofibrillar proteins to bind to the wells. Wells were then washed with washing buffer (HEPES 0.05 M, NaCl 0.15 M, Tween 20 0.05% (*v*/*v*), NaN_3_ 0.05% (*w*/*v*)). Plates were then loaded with a blocking buffer (0.05 M HEPES, 0.1% NaN_3_ (*w*/*v*), 3% non-fat dried milk (*w*/*v*)), and incubated at room temperature for 20 min. Wells were washed 3 times with washing buffer. Plates were incubated in the presence of primary monoclonal antibodies (type I = A4.840, type IIA = 6B8, and type IIB = BF-F3; Depreux et al. [24]) for 2 h at room temperature. Wells were then washed 3 times with a washing buffer, and a secondary alkaline phosphatase-conjugated anti-mouse immunoglobulin antibody was applied to each well. Plates were incubated for 90 min at room temperature. Wells were again washed 3 times with washing buffer, and AP substrate pNPP solution (KPL) was added. Plates were incubated for approximately 1 h until color developed, at which time 5% EDTA was added to stop the reaction. Absorbance was recorded at 410 nm using a Spectracount plate reader. Samples were normalized to standards on each plate to account for plate-to-plate variation.

### 2.7. Statistical Analyses

Prior to data analysis, normality of residuals and homogeneity of variances were tested using the Shapiro-Wilk and Levene tests, respectively, and outliers were removed. Data were analyzed by using the MIXED procedure of SAS software (9.4 SAS Institute Inc., Cary NC, USA), in a 3 × 2 × 2 factorial arrangement, considering the GP, ES voltage, ES time, and their interaction as fixed effects. Myosin heavy chain abundance was evaluated using the MIXED procedure, considering the genotypes as fixed effects. Muscle pH, temperature, and metabolite concentrations were analyzed as repeated measurements considering GP, ES voltage, ES time, time postmortem, and their interaction as fixed effects. Covariance structures were tested for each characteristic and the best fit was used. Pearson correlation coefficients were obtained using the CORR procedure to determine relationships between muscle myosin heavy chain isoform abundance, metabolites, and meat quality traits. Differences were considered statistically significant when *p* ≤ 0.05 and a tendency when 0.05 < *p* > 0.10.

## 3. Results

### 3.1. Myosin Heavy Chain Abundance (MyHC)

Myosin heavy chain isoform abundance was affected by genetic group (Table 2), where G2 animals had higher (*p* = 0.0311) relative abundance of type IIA isoform compared to G1 and G3; however, no differences were observed for type I, IIA/X, and type IIB between treatments.

### 3.2. Metabolites

No interactions of GP, ES voltage, time of ES, and time postmortem were observed for metabolites and glycolytic potential analysis. A tendency for genotype and time postmortem interaction was observed for G6P, where GP1 had higher concentrations at 1440 min compared to GP2 and GP3 (*p* = 0.0800; Table 3). Genotype affected metabolite concentration with higher glycogen in LL of GP3 animals (*p* = 0.018), whereas GP1 animals presented higher G6P (*p* = 0.0040) and lactate (*p* = 0.0003).

Higher glycolytic potential was observed in GP3 animals (*p* = 0.0369). Lactate was higher in carcasses stimulated with 200 V than 100 V (*p* = 0.0247; Table 4). No effect of ES time was observed for metabolites or glycolytic potential (Table 5).

### 3.3. Meat Quality Traits

No GP, ES voltage, and time of ES interactions were observed for meat quality traits (Table 6). No effect of GP or time of ES was observed for meat quality traits; however, carcasses stimulated with 200 V presented paler subjective color (*p* < 0.0001), lower firmness (*p* < 0.0001), and higher drip loss (*p* = 0.0020) than carcasses stimulated with 100 V. As expected, higher L* (*p* = 0.0066) and b* values (*p* = 0.0166) were also observed in carcasses stimulated with higher voltage.

### 3.4. pH and Temperature

Although no GP, ES voltage, time of ES, and time postmortem interactions were observed, the higher voltage accelerated the pH decline at 20 min up to 60 min postmortem (*p* < 0.0001; Figure 1b). Also, carcasses that were stimulated at 15 min postmortem showed lower pH up to 60 min postmortem when compared to carcasses stimulated at 25 min (*p* < 0.0001; Figure 1c). Carcasses from GP3 had higher overall pH values (*p* = 0.0003; Figure 1a). Also, a tendency for voltage, time of ES, and time postmortem interaction was observed for pH decline (*p* = 0.0933). Carcasses stimulated earlier with a higher voltage (200 V–15 min) showed a faster rate of pH decline up to 60 min postmortem; however, no differences in ultimate pH (pHu) were observed (Figure 1d).

No GP, ES voltage, time of ES, and time postmortem interactions were observed for temperature postmortem (Figure 2). Higher temperatures were observed in carcasses that were stimulated with 200 V than in carcasses stimulated with 100 V (Figure 2b). No effect of genetic groups or time of ES was observed for temperature decline.

### 3.5. Correlations

A strong positive correlation between type IIA and type IIA/X MyHC isoform abundance was observed (r = 0.56; *p* = 0.0489; Table 7). In addition, a strong negative correlation was observed between type IIA/X and type IIB MyHC abundance (r = −0.77; *p* < 0.01).

Fiber types, or MyHC isoforms, also had strong correlations with meat quality traits (Table 8). Type I MyHC isoform abundance is strongly correlated with meat color (r = 0.58; *p* = 0.0281), while type IIB MyHC abundance tended to be positively correlated with L* (r = 0.50; *p* = 0.0669) and negatively correlated with a* (r = −0.56; *p* = 0.0386).

In addition, type I MyHC isoform was negatively correlated with glycogen at 0, 30, 60, and 1440 min postmortem (*p* < 0.05; Table 9) and positively correlated with lactate at 0 and 30 min postmortem (*p* < 0.10), while type IIA MyHC isoform content in muscle was positively correlated with lactate and glucose at 30 and 60 min postmortem (*p* < 0.10).

Metabolites were moderately to lowly correlated to some meat quality traits (Table 10). Lactate at 30, 60, and 1440 min, and glucose at 60 min postmortem were positively correlated with L* (*p* < 0.050), while a* values were negatively correlated with lactate and glucose at 1440 min postmortem (*p* < 0.010). Glycogen at 1440 min postmortem was positively correlated with b* value. Subjective meat color was negatively correlated with lactate at 60 min, and G6P at 0, 1440 min postmortem (*p* < 0.050). Firmness and drip loss were negatively correlated with G6P at 60 and 1440 min postmortem (*p* < 0.050).

## 4. Discussion

The intent of this study was to understand better the role of genetic populations on postmortem energy metabolism and to explore further the development of adverse pork quality. To do this, ES was utilized in an attempt to determine if any of the three genetic populations were more susceptible to PSE or adverse pork quality events. During normal postmortem metabolism, the pH typically declines from 7.4, which is observed in living muscle, to an ultimate pHu of about 5.5 [25]. It is well established that any deviations from normal pH decline can adversely impact fresh meat quality [5,25,26,27,28,29]. Thus, the rate and extent of postmortem pH decline and carcass temperature are critical in the development of PSE meat [15,30,31]. Carcasses with an extended pH decline or low pHu of around 5.3 are hallmarks of acid pork, usually instigated by the presence of the RN gene [32]. The low pHu causes decreased water-holding capacity, thus leading to increased purge [33,34].

Although it is widely discussed how pH and postmortem glycolysis directly affects meat quality traits, there were no differences in quality traits between GP, even though there were differences in muscle pH and postmortem glycolysis among the genetic populations. Again, this argues that pH decline alone fails to explain global differences in meat quality development in pork carcasses [25,26].

The three genotypes showed a similar pattern of muscle MyHC isoform, as indicated by MyHC isoform abundance, apart from the relative abundance of type IIA, which was higher in GP2. The GP3 had higher pH compared to the other GPs and, conversely, a lower glycogen degradation and lower lactate accumulation. Interestingly, when we compared the abundance of the MyHC isoforms across GPs, we observed that even though not statistically significant, GP3 had a lower abundances of types IIA/X and IIB when compared to the other GPs. Skeletal muscle consists of a heterogeneous population of MyHC isoforms differing in their ability to contract and metabolize energy [35,36]. Muscle fibers exhibit remarkable dynamic adaptability, characterized by significant plasticity following the obligatory pathway I <-> IIa <-> IIx <-> IIb, leading to a change in their phenotype or type, directly impacting muscle energy metabolism. To date, four adult muscle MyHC isoforms I, IIA, IIX(D) and IIB, have been identified in pig skeletal muscle [37,38]. Paler, less red muscles have more glycolytic fibers (IIX and IIB) that are particularly susceptible to rapid postmortem glycolysis and PSE development [4,39,40]. This is because muscles containing predominantly glycolytic fibers (IIX and IIB) possess greater ATPase activity, a major driver of glycolytic flux postmortem [39,41]. Furthermore, calcium homeostasis differs with muscle fiber type [41], especially in halothane-positive genotypes [42]. While the latter genotypes are not present in animals used in this study, selection for faster growing, meatier animals increases the predominance of fast-contracting fiber types and whiter meat [43,44,45]. Similarly, correlation data showed a positive relationship between MyHC IIB isoform and L* values and a negative relationship with a* values.

The three genetic lines responded similarly to ES, but a comparison of ES treatment groups revealed a trend for three-way interaction between voltage, time of ES, and time postmortem. This trend suggests that the greater voltage with an earlier application results in a greater response of muscle pH decline. This is consistent with the work of Hamalman et al. [46], who titrated ES in pork carcasses and observed the greatest effect earlier postmortem. We have recently affirmed these findings and shown that even mechanical perturbation of carcasses aggravates muscle pH decline at earlier times in the process postmortem [47]. Consistent with this notion, carcasses stimulated with 200 V at 15 min postmortem had the fastest rate of pH decline. However, even with these differences, the pHu was similar between treatments, above 5.50. As Bowker et al. [4] stated, inconsistencies in the data can to some extent be explained by accounting for differences in both the level and time of ES, or level and magnitude of the assault to the carcass [47]. Regardless, the use of ES models to investigate the development of PSE has limitations. As shown in these data, high-voltage ES early postmortem only partially mimicked the development of the PSE condition, with a faster rate of pH decline observed early postmortem, yet the pHu remained normal. Others have reported that ES accelerates pH decline but has a minimal effect on pork quality characteristics [48,49,50]. Even so, we observed poor meat quality, specifically in high-voltage ES carcasses, even with a normal pHu.

Classically, an increased rate of postmortem muscle metabolism is marked by an increase in lactate production. Carcasses stimulated with 200 V, which tended to have the fastest rate of pH decline, showed higher lactate production but, interestingly, did not have differences in glycogen degradation or formation of G6P and glucose. Meat from carcasses stimulated with 200 V was paler with higher L* values; also, a moderate correlation was observed with L* and lactate at 30, 60, and 1440 min postmortem. However, muscle glycogen concentrations and glycolytic potential were higher in GP3 than in GP1, with GP2 having intermediate glycogen concentrations. At 1440 min postmortem, glycogen concentration was not different across GPs, with average concentrations of approximately 20 μmoles/g. The results of this study support the findings by Sayre et al. [11] and findings by Monin et al. [32], which illustrated that muscle metabolism can stop in the presence of residual glycogen. This is most likely due to the intricate metabolic properties of skeletal muscle beyond glycogen content that determine the culmination of lactate and subsequent pH decline [25]. To date, there are two feasible hypotheses, one being that there is a pH-mediated inactivation of glycolytic enzymes, which halts hydrogen accumulation at a constant endpoint, and the other being that there is a loss of adenosine nucleotides, preventing a glycolytic substrate undergo re-phosphorylation [43]. A limitation of the current study is that only four intermediates of glycolysis were measured. There are, however, nine metabolic intermediates between G6P and lactate, some of which are rate-limiting [26].

Muscle glycogen is metabolized into glucose and glucose-1-phosphate, the precursors of G6P by glycogen phosphorylase. In the present study, GP differences in G6P concentrations were observed for ES carcasses. Overall, concentrations of G6P decreased during the first hour postmortem regardless of GP and then increased at 1440 min, with GP1 having greater accumulations of G6P than GP2 and GP3. Depletion of G6P early indicates that glycogen breakdown may not be sufficient to supply glycolysis early postmortem. Previous studies had indicated a potential link between pHu and the proportion of glycogen phosphorylase. Scopes [51] suggested that the glycogen phosphorylase step is rate-limiting in postmortem metabolism [9,52,53]. However, Bowker et al. [4] concluded that glycogen phosphorylase is not likely the primary regulator of the rapid postmortem metabolism in PSE-prone muscles. Additionally, Morgan et al. [54], Kastenschmidt et al. [53] and Hamm [55] showed that ATP counteracts the activating influences of AMP and Pi, thus inhibiting glycogen phosphorylase activity. In addition, glucose is not being converted to G6P, further limiting the supply of G6P early postmortem. However, G6P concentrations increase in the latter stages of metabolism, indicating that glycogen phosphorylase activity is increasing or that the activity of enzymes “post-G6P”, such as phosphoglucose isomerase or phosphofructokinase (PFK), is decreasing. PFK irreversibly catalyzes the conversion of fructose-6-phosphate into fructose 1,6-bisphosphate and is considered the rate-limiting enzyme in the glycolytic pathway. It has been established that pH inactivates PFK and arrests postmortem glycolysis [4,26]. It is also unlikely that glycogen phosphorylase activity is increasing, because carcass temperature and muscle pH are declining during this time, making it unfeasible to increase the active amount of glycogen phosphorylase. This again supports studies that indicate that the inactivation of glycolysis is a result of enzymes downstream from G6P in glycolysis [32,53] rather than glycogen phosphorylase, thus making it more plausible that the inactivation of PFK late in postmortem glycolysis is responsible for a decreased rate of G6P metabolism and the subsequent backlog of G6P.

Glycolytic potential levels can only explain roughly 35 to 50% of the variation in pHu, suggesting glycolysis is controlled by substrate availability and enzyme activity [56,57]. The observations in this study are similar to reports of Bowker et al. [17] and Crenwelge et al. [58] that carcasses subjected to ES exhibit more rapid rates of pH decline than non-stimulated carcasses, potentially altering the flux through glycolysis and the amount of available glycogen utilized. Interestingly, we could not observe a faster rate of glycolysis in higher-voltage stimulated carcasses, except for an increased lactate accumulation. Indeed, GP3 possessed higher muscle pH than GP1 from 20–40 min postmortem when carcasses were stimulated, signifying a difference in the rate of pH decline once a challenge was applied to the muscle. Regardless, additional fundamental research is needed to understand precisely how glycolytic flux can be greater without having a corresponding decrease in tissue pH, and how lactate can increase without a subsequent decrease in glycogen, all in the absence of changes in pork quality development. To date, there is little biochemical evidence to support the uncoupling of glycolytic flux and pH decline in muscle.

## 5. Conclusions

Consistent with the notion that genotype influences meat quality, we showed herein that glycolytic metabolites differ with genetic line postmortem. However, we were unable to show differences in pork quality development with genetic line and ES. We were, however, able to show that higher voltages accelerated the rate of pH decline early postmortem, reducing pork quality attributes while not affecting glycogen degradation or glycolytic potential. Further investigations into the myriad of regulatory factors controlling postmortem energy metabolism are necessary for us to have a better understanding of how fresh meat quality is developed.

## Figures and Tables

**Figure 1 animals-13-02599-f001:**
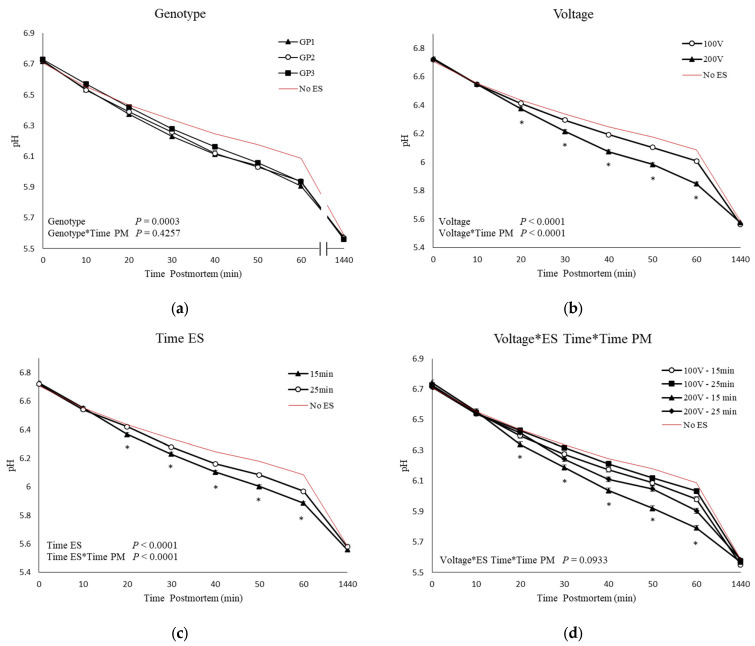
Means and SEM of Longissimus thoracis muscle pH values of three genetic populations (GP1-3; (**a**)) of stimulated carcasses (100 and 200 V; (**b**)), carcasses stimulated at different times postmortem (15 and 20 min; (**c**)); and carcasses stimulated with different voltages (100 and 200) at 15- and 25-min postmortem (**d**). Controls (no ES; lines without symbols) are included as a reference for non-stimulated carcasses but are not included in the statistical model. * *p* < 0.05.

**Figure 2 animals-13-02599-f002:**
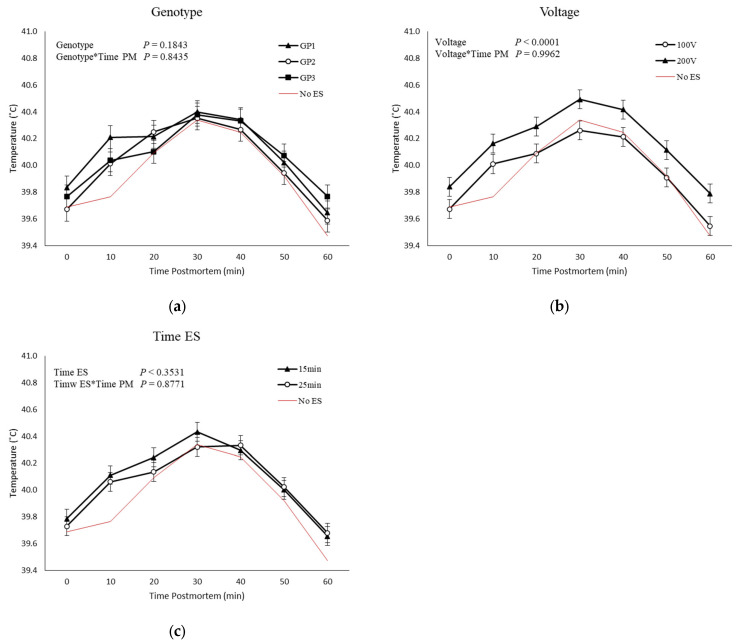
Means and SEM of Longissimus thoracis muscle temperatures of three genetic populations (GP1-3; (**a**)) of stimulated carcasses (100 and 200 V; (**b**)), carcasses stimulated at different time postmortem (15 and 20 min; (**c**)). Controls (no ES; lines without symbols) are included as a reference for non-stimulated carcasses but are not included in the statistical model.

**Table 1 animals-13-02599-t001:** Characterization of growth performance and carcass traits of three genetic populations (GP) of pigs.

GPs Characteristics	Genotypes		
	GP1	GP2	GP3
Start wt, kg ^b^	29.2 ± 0.40 ^x^	24.0 ± 0.40 ^y^	20.7 ± 0.40 ^z^
Slaughter wt., kg ^b^	112.7 ± 0.69 ^y^	114.3 ± 0.68 ^xy^	115.5 ± 0.69 ^x^
Slaughter age, day ^b^	167 ± 1.36 ^y^	178 ± 1.35 ^x^	166 ± 1.36 ^y^
Average daily gain, kg/day ^b^	0.73 ± 0.01 ^y^	0.75 ± 0.01 ^y^	0.83 ± 0.01 ^x^
10th rib LMA, cm^2 c^	44.8 ± 0.79 ^x^	43.2 ± 0.88 ^y^	45.9 ± 0.77 ^x^
10th rib backfat, mm ^c^	18.8 ± 0.64 ^y^	22.6 ± 0.72 ^x^	23.2 ± 0.63 ^x^
Last rib fat depth, mm ^c^	22.9 ± 0.93 ^y^	24.6 ± 1.0 ^x^	26.2 ± 0.91 ^x^
Calculated % lean ^c^	54.7 ± 0.43 ^x^	52.7 ± 0.48 ^y^	53.3 ± 0.42 ^y^

GP1 1; n = 49; GP2 2, n = 50; GP3 3, n = 49. ^b^ GL 1; n = 41; GL 2, n = 32; GL 3, n = 42. ^c^ NPPC, 1991. ^xyz^ Means lacking a common superscript differ (*p* < 0.05).

**Table 2 animals-13-02599-t002:** Relative abundance (LSM ± SE) of type I, IIA, IIA/X and IIB myosin heavy chain (MyHC) isoforms in the longissimus muscle of pigs from three genetic populations (GP).

MyHC Isoforms	Genotypes			*p* Value
	GP1 (n = 10)	GP2 (n = 10)	GP3 (n = 10)	
Type I	0.62 ± 0.03	0.56 ± 0.03	0.62 ± 0.03	0.2834
Type IIA	0.53 ± 0.03 ^y^	0.64 ± 0.03 ^x^	0.53 ± 0.03 ^y^	0.0311
Type IIA/X	0.35 ± 0.15	0.77 ± 0.15	0.46 ± 0.15	0.1638
Type IIB	1.6 ± 0.10	1.4 ± 0.10	1.4 ± 0.10	0.2665

^xy^ Means lacking a common superscript differ (*p* < 0.05).

**Table 3 animals-13-02599-t003:** Glycogen, glucose, glucose 6-phosphate (G6P), lactate and glycolytic potential of muscle from electrically stimulated and non-stimulated carcasses across two time points postmortem (PM) from three genetic populations (GP) of pigs taken at 1, 30, 60 and 1440 min PM.

Traits (µmol/g)	Time PM (min)	Genotypes			*p* Value		
		GP1 (n = 20)	GP2 (n = 20)	GP3 (n = 20)	Genotype	Time PM	Gen × Time PM
Glycogen	1	55.7 ± 4.85	57.5 ± 5.00	63.6 ± 4.85	0.0185	<0.0001	0.7957
	30	51.8 ± 4.85	55.4 ± 4.85	64.3 ± 4.85			
	60	42.2 ± 4.85	51.6 ± 4.85	59.2 ± 4.85			
	1440	20.3 ± 4.85	20.2 ± 4.85	21.6 ± 4.85			
Glucose	1	3.9 ± 0.34	3.9 ± 0.35	3.7 ± 0.34	0.7435	<0.0001	0.7524
	30	4.3 ± 0.34	4.3 ± 0.34	4.1 ± 0.34			
	60	5.2 ± 0.34	4.8 ± 0.34	4.4 ± 0.34			
	1440	7.6 ± 0.34	7.5 ± 0.34	8.0 ± 0.34			
G6P	1	11.9 ± 0.49	11.9 ± 0.49	10.9 ± 0.49	0.0040	<0.0001	0.0800
	30	9.7 ± 0.50	9.2 ± 0.49	9.3 ± 0.49			
	60	7.4 ± 0.49	7.3 ± 0.49	6.8 ± 0.49			
	1440	11.4 ± 0.50	8.8 ± 0.49	8.8 ± 0.50			
Lactate	1	43.7 ± 3.08	40.2 ± 3.18	38.0 ± 3.08	0.0003	<0.0001	0.2480
	30	57.4 ± 3.08	52.6 ± 3.18	48.2 ± 3.08			
	60	78.8 ± 3.08	64.1 ± 3.18	61.4 ± 3.18			
	1440	109.6 ± 3.08	108.6 ± 3.08	106.6 ± 3.08			
Glycolytic Potential	1	186.9 ± 8.04	186.9 ± 8.29	200.8 ± 8.29	0.0369	<0.0001	0.7425
	30	188.0 ± 8.04	194.1 ± 8.04	203.6 ± 8.04			
	60	188.2 ± 8.04	179.6 ± 8.29	205.4 ± 8.04			
	1440	188.3 ± 8.04	176.5 ± 8.29	184.5 ± 8.04			

**Table 4 animals-13-02599-t004:** Glycogen, glucose, glucose 6-phosphate (G6P), lactate and glycolytic potential of muscle from electrically stimulated carcasses with 100 or 200 volts (V) from three genetic populations (GP) of pigs taken at 1, 30, 60 and 1440 min postmortem (PM).

Traits (µmol/g)	Time PM (min)	Voltages		*p* Value		
		100 V (n = 30)	200 V (n = 30)	Voltage	Time PM	Voltage × Time PM
Glycogen	1	57.7 ± 4.04	60.2 ± 3.96	0.3264	<0.0001	0.4074
	30	61.0 ± 3.96	53.3 ± 3.96			
	60	54.7 ± 3.96	47.3 ± 3.96			
	1440	20.0 ± 3.96	21.5 ± 3.96			
Glucose	1	4.0 ± 0.28	3.7 ± 0.28	0.5644	<0.0001	0.6380
	30	4.4 ± 0.28	4.1 ± 0.28			
	60	4.9 ± 0.28	4.7 ± 0.28			
	1440	7.5 ± 0.28	7.9 ± 0.28			
G6P	1	11.8 ± 0.41	11.4 ± 0.40	0.8359	<0.0001	0.4997
	30	9.0 ± 0.41	9.7 ± 0.40			
	60	7.3 ± 0.40	7.0 ± 0.40			
	1440	9.6 ± 0.40	9.9 ± 0.41			
Lactate	1	38.2 ± 2.57	43.1 ± 2.52	0.0247	<0.0001	0.9131
	30	49.9 ± 2.57	55.6 ± 2.52			
	60	66.7 ± 2.57	69.5 ± 2.57			
	1440	106.9 ± 2.52	109.7 ± 2.52			
Glycolytic Potential	1	189.3 ± 6.83	193.7 ± 6.60	0.9209	<0.0001	0.3910
	30	200.5 ± 6.60	189.9 ± 6.60			
	60	192.6 ± 6.70	189.5 ± 6.60			
	1440	177.5 ± 6.70	188.6 ± 6.60			

**Table 5 animals-13-02599-t005:** Glycogen, glucose, glucose 6-phosphate (G6P), lactate and glycolytic potential of muscle from electrically stimulated carcasses across 100 and 200 volts (V) at 15 and 25 min postmortem (PM) from three genetic populations (GP) of pigs taken at 1, 30, 60 and 1440 min PM.

Traits (µmol/g)	Time PM (min)	Voltages		*p* Value		
		15 min (n = 30)	25 min (n = 30)	Time ES	Time PM	Time ES × Time PM
Glycogen	1	58.0 ± 4.04	59.8 ± 3.96	0.7788	<0.0001	0.8803
	30	58.6 ± 3.96	55.7 ± 3.96			
	60	49.4 ± 3.96	52.6 ± 3.96			
	1440	20.2 ± 3.96	21.2 ± 3.96			
Glucose	1	3.7 ± 0.28	4.0 ± 0.28	0.6144	<0.0001	0.5242
	30	4.3 ± 0.28	4.2 ± 0.28			
	60	4.8 ± 0.28	4.8 ± 0.28			
	1440	8.0 ± 0.28	7.4 ± 0.28			
G6P	1	11.4 ± 0.41	11.7 ± 0.40	0.7729	<0.0001	0.8610
	30	9.2 ± 0.41	9.6 ± 0.40			
	60	7.1 ± 0.40	7.1 ± 0.40			
	1440	9.8 ± 0.41	9.6 ± 0.40			
Lactate	1	41.5 ± 2.57	39.7 ± 2.52	0.6764	<0.0001	0.8604
	30	52.2 ± 2.57	53.3 ± 2.52			
	60	66.8 ± 2.57	69.3 ± 2.57			
	1440	107.7 ± 2.52	108.8 ± 2.52			
Glycolytic Potential	1	192.2 ± 6.83	190.9 ± 6.57	0.5647	<0.0001	0.9817
	30	198.2 ± 6.57	192.3 ± 6.57			
	60	191.6 ± 6.57	190.5 ± 6.70			
	1440	184.3 ± 6.57	181.8 ± 7.70			

**Table 6 animals-13-02599-t006:** Quality traits (LSM ± SE) of electrically stimulated (ES) carcasses from three genetic populations (GP) of pigs, across two voltages (V; 100 and 200 V), and time of ES (15 and 25 min).

Traits (µmol/g)	Genotypes			Voltage		ES Time		*p* Value		
	GP1	GP2	GP3	100 V	200 V	15 min	25 min	Genotype	Voltage	Time ES
L*	64.4 ± 0.44	63.3 ± 0.45	63.9 ± 0.45	63.1 ± 0.36	64.6 ± 0.36	63.7 ± 0.36	64.0 ± 0.36	0.2382	0.0066	0.6028
a*	12.6 ± 0.23	12.9 ± 0.23	12.8 ± 0.23	12.6 ± 0.19	12.9 ± 0.19	12.8 ± 0.19	12.7 ± 0.19	0.7622	0.3072	0.5988
b*	11.2 ± 0.20	11.1 ± 0.20	11.4 ± 0.20	10.9 ± 0.17	11.5 ± 0.17	11.2 ± 0.17	11.2 ± 0.17	0.5015	0.0166	0.8013
Color ^a^	1.9 ± 0.07	2.0 ± 0.07	2.0 ± 0.07	2.2 ± 0.05	1.8 ± 0.05	1.9 ± 0.05	2.0 ± 0.05	0.6156	<0.0001	0.1236
Firmness ^b^	1.7 ± 0.07	1.8 ± 0.07	1.9 ± 0.07	2.0 ± 0.06	1.6 ± 0.06	1.8 ± 0.06	1.9 ± 0.06	0.3955	<0.0001	0.1897
Drip Loss, %	6.5 ± 0.29	6.9 ± 0.29	6.5 ± 0.29	6.1 ± 0.23	7.2 ± 0.24	6.8 ± 0.23	6.4 ± 0.24	0.5351	0.0020	0.2520

L* = Lightness; a* = redness; b* = yellowness; ^a^ Color score of 1 = pale pinkish gray to white, 5 = dark purplish red; ^b^ Firmness score of 1 = very soft, 5 = very firm [19]. % Drip Loss = percent drip loss [21].

**Table 7 animals-13-02599-t007:** Correlation coefficients among muscle myosin heavy chain (MyHC) isoforms (Type I, IIA, IIA/X and IIB), in *Longissimus* muscle.

MyHC	Type IIA	Type IIA/X	Type IIB
Type I	−0.14	−0.40	0.33
Type IIA		0.56 **	0.26
Type IIA/X			−0.77 ***

** *p* < 0.05, *** *p* < 0.01.

**Table 8 animals-13-02599-t008:** Correlation coefficients between myosin heavy chain isoform (MyHC) isoforms (Type I, IIA, IIA/X and IIB), pH and meat quality traits.

MyHC	pH 0	pH 10	pH 20	pH 30	pH 40	pH 50	pH 60	pH 1440	L*	a*	b*	Color	Firmness	DL
Type I	−0.40	−0.04	−0.11	−0.28	−0.13	−0.04	0.20	−0.11	0.20	0.04	0.38	0.58 **	−0.13	−0.36
Type IIA	−0.18	0.021	−0.41	−0.02	−0.06	−0.02	0.22	−0.11	0.04	−0.32	−0.26	−0.09	−0.40	−0.23
Type IIA/X	0.02	0.03	−0.15	0.08	0.07	0.21	0.29	0.09	−0.38	0.22	−0.35	−0.11	0.08	0.03
Type IIB	−0.05	0.11	−0.12	−0.10	−0.11	−0.26	−0.20	−0.07	0.50 *	−0.56 **	0.09	0.02	−0.37	−0.30

Carcass pH measured at times (min) post-exsanguination; L* = luminosity; a* = redness; b* = yellowness; Color score of 1 = pale pinkish gray to white, 5 = dark purplish red; Firmness score of 1 = very soft, 5 = very firm [19]; % Drip Loss = percent drip loss [21]. * *p* < 0.10, ** *p* < 0.05.

**Table 9 animals-13-02599-t009:** Correlation coefficients between myosin heavy chain isoform (MyHC) isoforms (Type I, IIA, IIA/X and IIB), and metabolite concentration.

MyHC	Lac 0	Lac 30	Lac 60	Lac 1440	G6P 0	G6P 30	G6P 60	G6P 1440	Glu 0	Glu 30	Glu 60	Glu 1440	Gly 0	Gly 30	Gly 60	Gly 1440
Type I	0.64 **	0.53 *	0.02	0.10	0.14	0.35	0.12	−0.65 **	0.25	0.49 *	0.22	−0.65 **	−0.70 ***	−0.59 **	−0.62 **	−0.55 **
Type IIA	0.40	0.50 *	0.68 **	−0.02	0.09	−0.02	0.12	−0.22	0.69 ***	0.40	0.56 **	0.05	−0.22	−0.35	−0.47	−0.45
Type IIA/X	−0.12	0.09	0.21	−0.22	−0.20	−0.08	−0.28	0.06	0.17	0.16	0.13	0.39	0.24	0.15	0.31	0.23
Type IIB	0.27	0.34	0.20	0.15	0.22	−0.06	0.31	−0.31	0.28	0.29	−0.50 *	−0.36	−0.36	−0.35	−0.58	−0.51

Carcass metabolites measured at times (min) post-exsanguination; Lac = lactate; G6P = glucose-6-phosphate; Glu = glucose; Gly = glycogen. * *p* < 0.10, ** *p* < 0.05, *** *p* < 0.01.

**Table 10 animals-13-02599-t010:** Quality traits (LSM ± SE) of electrically stimulated carcasses from three genetic populations (GP) of pigs.

Quality traits	Lac 0	Lac 30	Lac 60	Lac 1440	G6P 0	G6P 30	G6P 60	G6P 1440	Glu 0	Glu 30	Glu 60	Glu 1440	Gly 0	Gly 30	Gly 60	Gly 1440
L*	0.17	0.34 ***	0.36 ***	0.27 **	0.18	0.11	0.24 **	0.09	−0.06	−0.06	0.25 **	−0.02	−0.09	−0.16	−0.14	−0.04
a*	−0.09	−0.10	−0.07	−0.29 **	−0.11	0.01	−0.03	−0.16	0.07	0.19	0.08	−0.20 *	−0.19	−0.15	−0.09	−0.08
b*	0.04	0.01	−0.08	−0.04	−0.06	0.10	−0.12	0.10	0.18	0.14	−0.16	0.02	0.08	0.09	0.19	0.23 **
Color	0.01	−0.14	−0.24 **	−0.12	−0.23 **	−0.23 *	−0.11	−0.25 **	0.12	0.02	−0.18	−0.11	−0.08	−0.01	0.05	−0.11
Firmness	−0.19	−0.22 *	−0.19 *	−0.21 *	−0.29 **	−0.03	−0.26 **	−0.24 **	−0.05	0.02	−0.10	−0.06	0.06	−0.02	0.02	−0.06
DL, %	0.17	0.20 *	0.29 **	0.18	0.20 *	0.09	0.31 ***	0.26 **	−0.17	−0.02	0.14	0.15	0.13	0.16	0.08	0.05

Carcass metabolites measured at times (min) post-exsanguination; Lac = lactate; G6P = glucose-6-phosphate; Glu = glucose; Gly = glycogen. L* = luminosity; a* = redness; b* = yellowness; Color score of 1 = pale pinkish gray to white, 5 = dark purplish red; Firmness score of 1 = very soft, 5 = very firm [19]; % Drip Loss = percent drip loss [21]. * *p* < 0.10, ** *p* < 0.05, *** *p* < 0.01.

## Data Availability

The data that support the findings of this study are available from the corresponding author, D.E.G., upon request.

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
