# Peer review of "Postmortem Metabolism and Pork Quality Development Are Affected by Electrical Stimulation across Three Genetic Lines"

_animals, 2023, doi:10.3390/ani13162599_

Round 1

Reviewer 1 Report

The research done is not innovative, but quite interesting. They are of practical importance.

In my opinion, the authors should make some minor revision.

The title is not appropriate for this manuscript – it is currently appropriate for a review, not a research, manuscript.

The purpose of the research was presented correctly. However, Materials and Methods need to be supplemented.

Information on the sex of the fatteners in the experiment should be provided. They should also be included in the results.

Feeding and housing conditions affect the quality of pork - this information should be supplemented in the manuscript.

The other components of the manuscript are acceptable.

Author Response

Reviwer1

The title is not appropriate for this manuscript – it is currently appropriate for a review, not a research, manuscript.

We edited the title according to the recommendations.

The purpose of the research was presented correctly. However, Materials and Methods need to be supplemented.

We improved the material and methods according to the recommendations.

Information on the sex of the fatteners in the experiment should be provided. They should also be included in the results.

We improved the material and methods according to the recommendations.

Feeding and housing conditions affect the quality of pork - this information should be supplemented in the manuscript.

We improved the material and methods according to the recommendations.

Reviewer 2 Report

The manuscript presents postmortem metabolism and pork quality development in three different pig breeds subjected to ES in order to stimulate adverse handling events after stunning. Generally, the research was well conceived and well presented. The paper provide a good review of mechanisms controlling biochemistry of muscles after the slaughter. The statistical analysis, also seems appropriate for this kind of research. The ES approach in order to accelerate of post mortem glycolysis have already been used in previous studies by other authors. Although, the advantage of this paper are genetic populations used in research. Modern pig lines and their crosses react differently on various pre- and post- slaughter factors exhibiting different patterns of post mortem metabolism in comparison to previous populations. However, there are few corrections that need to be done to improve the quality of the manuscript. Thus, I recommend the acceptance of the manuscript for publication subject to a minor revision. The corrections are as follows:

 19-20 “The three lines differed in their response to electrical stimulation, specifically accumulated greater amounts lactate but breakdown of glycogen unchanged” –  Which of these genetic populations respond differently and for which ES? 100V at 15 min? 100V at 25min? 200V at 15 min? or 200V at 25min?. Different lactate concentrations from carcasses stimulated 100 and 200V have been presented in table 3 but regardless of genetic population. In turn, table 2 presents concentration of metabolites between genetic populations but with average records pooled from electrically stimulated and non-stimulated carcasses across two time points postmortem. Thus, such conclusion with reference to presented results is overinterpretation.  

305 Type IIA/X or IIA?

317-318 negative relationship between the abundance of type I MyHC and a* values? check it

346-348 “…20 μmoles/g. The results of this study support the findings by Sayre et al. [11] and findings by Monin et al. [32], which illustrated that muscle metabolism can stop in the presence of residual glycogen” Does it mean that there has been cessation of postmortem glycolysis? Did you analyse metabolites after 24h post mortem? Note that high glycolytic potential levels in muscles (above 180-190 μmoles/g) usually promotes extended postmortem glycolysis. Furthermore, in this research, the average levels of glycogen were approximately 20 μmoles/g. Maybe they are not so high as in RN- phenotype although such muscles usually utilise another 10-12 μmoles/g glycogen in the next 24h.

434 provide pages

438 provide journal and pages

445 names are in capital letters, change it

454 the title in capital letters, change it

480 provide journal and pages

Author Response

Reviewer 2

19-20 “The three lines differed in their response to electrical stimulation, specifically accumulated greater amounts lactate but breakdown of glycogen unchanged” –  Which of these genetic populations respond differently and for which ES? 100V at 15 min? 100V at 25min? 200V at 15 min? or 200V at 25min?. Different lactate concentrations from carcasses stimulated 100 and 200V have been presented in table 3 but regardless of genetic population. In turn, table 2 presents concentration of metabolites between genetic populations but with average records pooled from electrically stimulated and non-stimulated carcasses across two time points postmortem. Thus, such conclusion with reference to presented results is overinterpretation.

We edited the text according to the recommendations.

305 Type IIA/X or IIA?

We edited the text according to the recommendations.

317-318 negative relationship between the abundance of type I MyHC and a* values? check it

We edited the text according to the recommendations.

346-348 “…20 μmoles/g. The results of this study support the findings by Sayre et al. [11] and findings by Monin et al. [32], which illustrated that muscle metabolism can stop in the presence of residual glycogen” Does it mean that there has been cessation of postmortem glycolysis? Did you analyse metabolites after 24h post mortem? Note that high glycolytic potential levels in muscles (above 180-190 μmoles/g) usually promotes extended postmortem glycolysis. Furthermore, in this research, the average levels of glycogen were approximately 20 μmoles/g. Maybe they are not so high as in RN- phenotype although such muscles usually utilise another 10-12 μmoles/g glycogen in the next 24h.

We understand your point, however, when we study pH decline in non-stimulated pork carcasses, which rigor mortis is stablished around 4 - 6 hours post-exsanguination, we can speculate that at 24h postmortem, carcasses already reached their ultimate pH. Different than beef carcasses, that reach rigor around 12 hours post exsanguination, and sometimes, need more time to reach ultimate pH, maybe up to 48h.

When we have glycogen at 24h (residual glycogen), like in this study, we assume that there was a cessation of postmortem glycolysis. Although we did not measure pH and metabolites after 24h, we didn’t expect to see significant changes in these values. Specifically, because the carcasses utilized in this study were electrically stimulated, accelerating even more postmortem metabolism.

434 provide pages

We edited the references according to the recommendations.

438 provide journal and pages

We edited the references according to the recommendations.

445 names are in capital letters, change it

We edited the references according to the recommendations.

454 the title in capital letters, change it

We edited the references according to the recommendations.

480 provide journal and pages

We edited the references according to the recommendations.

Reviewer 3 Report

The work attempted to further clarify postmortem mechanisms of conversion of muscle to meat by inducing the electrical stimulation in three genetic lines of pigs. The work is thoroughly developed and there are few issues or comments.

General comments:

One general comment relates to the use of the term "pork quality." The paper focuses mainly on an aspect of pork quality called "technological quality" so I would advise being more specific and emphasizing this aspect, thereby avoiding to refer to pork quality in its wider sense.

Another general comment relates to the use of the term "muscle fiber type", since it was the abundance of the different types of myosin heavy chains that was determined, I would suggest that the term be rephrased to make it more precise.

The information on the conformation and leanness of the genetic lines used is missing. In addition, as the genetic lines were not very divergent with respect to the profile of myosin heavy chains, it would be good to povide the mentioned data and rationale on the choice of genetic lines.

Please explain also the discrepany between number of pigs starting the experiment (n=150 i.e. probably about 50 pigs per genetic line) and experimental units in analysis (Table 1, Table 2) i.e. How was the selection of pigs to be used for the study made?

Please check also the following minor issues:

- simple summary (line 18-19): differences in the abundance of different myosin heavy chains ; across genetic lines, ...

- line 31: rephrase "genetic line altered the ..." to "genetic line affected..."; not muscle fiber type composition but relative abundance of different myosin heavy chains

- line 56-58 the sentence is not clear, please rephrase.

- line 72: a bit unclear - perhaps rephrasing to be more clear that it is the combination of high T and fast pH decline...

- lines 98-103: it is unusual to see three-way crosses to be used as parental lines. Please provide some explanation on the choice of lines and how these lines differ with respect to lean meat content and conformation. Preferrably in the results (Table 1). Provide please the information one body weight and leanness for each genetic line.

lines 115-116: change 1440 min to 24h

lines 130-133: the information is missing about when the color measurements were taken. The information is also missing regarding the visual assessment of color and firmness (which appears later in the table)

line 154: please be precise and use the term relative abundance of myosin heavy chains, since the fiber types were not analysed

line 194: idem, use the term relative abundance of myosin heavy chains

- Table 6 has little relevance for the study (its rationale) and could be omitted

In Table 9 DL,% is used which has not been defined previously. Please change to drip loss, %

- line 302 for clarity sake, please rephrase: Although the genetic lines differed in muscle pH decline and postmortem glycolysis, this did not result in any notable meat quality differences.

Quality of English is excellent apart from some awkward/unclear sentences (at least to this reviewer).

Author Response

Reviewer 3

One general comment relates to the use of the term "pork quality." The paper focuses mainly on an aspect of pork quality called "technological quality" so I would advise being more specific and emphasizing this aspect, thereby avoiding to refer to pork quality in its wider sense.

We understand your comment, but we use the electrical stimulation as a tool to mimic the PSE condition in those lines. The focus wasn’t to study the effect of electrical stimulation on pork quality, rather if this tool would allow us to simulate the aberrant condition. So, we are respectfully, maintaining the term “meat quality”.

Another general comment relates to the use of the term "muscle fiber type", since it was the abundance of the different types of myosin heavy chains that was determined, I would suggest that the term be rephrased to make it more precise.

We edited the text according to the recommendations.

The information on the conformation and leanness of the genetic lines used is missing. In addition, as the genetic lines were not very divergent with respect to the profile of myosin heavy chains, it would be good to povide the mentioned data and rationale on the choice of genetic lines.

We edited the text according to the recommendations.

Please explain also the discrepany between number of pigs starting the experiment (n=150 i.e. probably about 50 pigs per genetic line) and experimental units in analysis (Table 1, Table 2) i.e. How was the selection of pigs to be used for the study made?

We edited the text according to the recommendations.

- simple summary (line 18-19): differences in the abundance of different myosin heavy chains ; across genetic lines, ...

We edited the text according to the recommendations.

- line 31: rephrase "genetic line altered the ..." to "genetic line affected..."; not muscle fiber type composition but relative abundance of different myosin heavy chains

We edited the text according to the recommendations.

- line 56-58 the sentence is not clear, please rephrase.

We edited the text according to the recommendations.

- line 72: a bit unclear - perhaps rephrasing to be more clear that it is the combination of high T and fast pH decline...

We edited the text according to the recommendations.

- lines 98-103: it is unusual to see three-way crosses to be used as parental lines. Please provide some explanation on the choice of lines and how these lines differ with respect to lean meat content and conformation. Preferrably in the results (Table 1). Provide please the information one body weight and leanness for each genetic line.

We edited the text and added more information according to the recommendations.

lines 115-116: change 1440 min to 24h

We understand the recommendation for changing to 24h, but to keep it consistent with the rest of the data presented in the paper, we are, respectfully, keeping it in minutes.

lines 130-133: the information is missing about when the color measurements were taken. The information is also missing regarding the visual assessment of color and firmness (which appears later in the table)

We had the reference for these analysis in the material and methods (line 121, ref [19]), but we added a brief description in the text according to the recommendations.

line 154: please be precise and use the term relative abundance of myosin heavy chains, since the fiber types were not analysed

We edited the text according to the recommendations.

line 194: idem, use the term relative abundance of myosin heavy chains

We edited the text according to the recommendations.

- Table 6 has little relevance for the study (its rationale) and could be omitted

We edited the text bringing the importance of the table.

In Table 9 DL,% is used which has not been defined previously. Please change to drip loss, %

We added the abbreviation in the material and methods, but we also have it in the table legend.

- line 302 for clarity sake, please rephrase: Although the genetic lines differed in muscle pH decline and postmortem glycolysis, this did not result in any notable meat quality differences.

We edited the text according to the recommendations.